# Synthesis and Antiproliferative Potential of Thiazole and 4-Thiazolidinone Containing Motifs as Dual Inhibitors of EGFR and BRAF^V600E^

**DOI:** 10.3390/molecules28247951

**Published:** 2023-12-05

**Authors:** Alaa A. Hassan, Nasr K. Mohamed, Ashraf A. Aly, Mohamed Ramadan, Hesham A. M. Gomaa, Ahmed T. Abdel-Aziz, Bahaa G. M. Youssif, Stefan Bräse, Olaf Fuhr

**Affiliations:** 1Chemistry Department, Faculty of Science, Organic Division, Minia University, Minia 61519, Egyptashrafaly63@yahoo.com (A.A.A.);; 2Pharmaceutical Organic Chemistry Department, Faculty of Pharmacy, Al-Azhar University, Assiut 71524, Egypt; elbashamohammed@yahoo.com; 3Department of Pharmacology, College of Pharmacy, Jouf University, Sakaka 72341, Aljouf, Saudi Arabia; hasoliman@ju.edu.sa; 4Pharmaceutical Organic Chemistry Department, Faculty of Pharmacy, Assiut University, Assiut 71526, Egypt; 5Institute of Organic Chemistry, Karlsruher Institut fur Technologie, 76131 Karlsruhe, Germany; 6Institute Karlsruhe of Nanotechnology (INT) and Karlsruhe Nano Micro Facility (KNMFi), Institute of Technology (KIT), 76344 Eggenstein-Leopoldshafen, Germany; olaf.fuhr@kit.edu

**Keywords:** thiazole, 4-thiazolidinone, EGFR, BRAF, dual inhibitors, anticancer

## Abstract

Thiazole and thiazolidinone recur in a wide range of biologically active compounds that reach different targets within the context of tumors and represent a promising starting point to access potential candidates for treating metastatic cancer. Therefore, searching for new lead compounds that show the highest anticancer potency with the fewest adverse effects is a major drug-discovery challenge. Because the thiazole ring is present in dasatinib, which is currently used in anticancer therapy, it is important to highlight the ring. In this study, cycloalkylidenehydrazinecarbothioamides (cyclopentyl, cyclohexyl, cyclooctyl, dihydronapthalenylidene, flurine-9-ylidene, and indolinonyl) reacted with 2-bromoacetophenone and diethylacetylenedicarboxylate to yield thiazole and 4-thiazolidinone derivatives. The structure of the products was confirmed by using infrared (IR) spectroscopy, nuclear magnetic resonance (NMR) spectroscopy, mass spectrometry, and single-crystal X-ray analyses. The antiproliferative activity of the newly synthesized compounds was evaluated. The most effective inhibitory compounds were further tested in vitro against both epidermal growth factor receptor (EGFR) and B-Raf proto-oncogene, serine/threonine kinase (BRAF^V600E^) targets. Additionally, molecular docking analysis examined how these molecules bind to the active sites of EGFR and BRAF^V600E^.

## 1. Introduction

Cancer is a significant global health challenge, impacting the lives of millions of people worldwide [1,2]. Chemotherapy remains the prevailing method for cancer treatment in current medical practice. While it offers advantages in addressing cancer, it also carries undesirable side effects that are attributable to its indiscriminate impact on healthy and cancerous cells [3,4]. Combination chemotherapy is one way to simultaneously block two or more targets. However, the pharmacokinetic profiles and metabolic stabilities of two or more drugs are often different. In addition, combination chemotherapy may result in risky drug–drug interactions [5,6]. Developing potent and less toxic anticancer chemotherapeutic agents can be achieved effectively by targeting multiple integrated signaling functions with a single molecule [7,8]. Kinases have been shown to play an important role in regulating many fundamental cancer processes, including tumor growth, metastasis, neovascularization, and chemotherapy resistance. As a result, kinase inhibitors have become a major focus of drug development, with several kinase inhibitors now receiving approval by the United States Food and Drug Administration (FDA) for various cancer indications [9,10].

The epidermal growth factor receptor (EGFR) and B-Raf proto-oncogene, serine/threonine kinase (BRAF) are both well-studied kinases that play crucial roles in cancer progression [11,12]. Unlike single-target inhibitors, dual inhibitors targeting both the EGFR and BRAF have the potential to provide greater efficacy and to overcome resistance. Combining EGFR and BRAF inhibition has resulted in synergistic antitumor effects in preclinical studies [13]. Furthermore, dual inhibitors may help prevent the development of resistance, a common problem with single-target inhibitors [14]. Therefore, developing dual EGFR/BRAF inhibitors is a promising approach in cancer therapy [15]. 

Thiazole and 4-thiazolidinone containing heterocyclic compounds, with a broad spectrum of pharmaceutical activities, represent a significant medicinal chemistry class. Thiazole and 4-thiazolidinone are five-membered unique heterocyclic motifs containing S and N atoms as the essential core scaffold, and they have commendable medicinal significance [16,17,18]. Thiazoles and 4-thiazolidinones containing heterocyclic compounds are building blocks for the next generation of pharmaceuticals. Thiazole precursors have been frequently used, due to their capability to bind to numerous cancer-specific protein targets. Suitably, thiazole motifs have a biological suit via inhibition of different signaling pathways involved in cancer causes. The scientific community has always tried to synthesize novel thiazole-based heterocycles by replacing functional groups or skeletons around the thiazole moiety [16]. 

The thiazole nucleus is a fundamental part of some clinically applied anticancer drugs, such as dasatinib (**1**) [17] and dabrafenib (**2**) [19] (Figure 1). Recently, thiazole-containing compounds have been successfully developed as possible inhibitors of several biological targets, including enzyme-linked receptor(s) located on the cell membrane (i.e., polymerase inhibitors) and the cell cycle (i.e., microtubular inhibitors) [20]. 

Aly et al. [21] synthesized and characterized six paracyclophanyl thiazolidinone-based compounds, which were subsequently screened against 60 different cancer cell lines. Compound **3** (Figure 2) showed comparatively better anticancer activity, overall, than the rest of the series, especially against two cell lines of leukemia (RPMI-8226 and SR). All the compounds exhibited anti-proliferative activity against RPMI-8226 and SR, with IC_50_ values of 1.61 µM and 1.11 µM, respectively. LV et al. [22] introduced two series of thiazolidinone derivatives and assayed for inhibitory action against EGFR and HER-2 kinases. Some of the synthesized compounds displayed potent EGFR and HER-2 inhibitory activities. Compound **4** displayed the most potent EGFR and HER-2 inhibitory activities (IC_50_; 0.09 µM for EGFR and IC_50_; 0.42 µM for HER-2) in MCF-7 cell lines, compared to erlotinib. Thiazolidin-4-one hybrids were developed by Aziz et al., and their anticancer properties were tested on breast cancer (MCF-7) and lung cancer (A549) cell lines. The most effective derivative against the lung cancer (A549) cell line was compound **5** (Figure 2), with an IC_50_ value of 0.72 µM and promising EGFR inhibitory activity at a concentration of 65 nM [23]. Several 4-(5-methylisoxazol-3-ylamino) thiazoles were prepared and evaluated as cytotoxic agents against three human cancer cell lines (HCT-116, HePG-2, and MCF-7). Compound **6** (Figure 2), with IC_50_ = 20.2 μg/mL against the Hep-G2 cell line, was as potent as the reference drug, doxorubicin (IC_50_ = 21.6 μg/mL). The in vitro kinases inhibitory assay revealed that this compound strongly inhibited 3 out of 12 kinases (EGFR, PI3K [p110b/p85a] and p38α) by 95%, 89%, and 85%, respectively. In addition, moderate-to-weak inhibitory activities were observed for the rest of kinases (AKT2, CDK2/Cylin A1, PDGFRβ, VEGFR-2, BRAF[V600E], CHK1, PI3K [p110a/p85a], c-RAF, and AKT1; inhibitions of 2–64%) [24].

Herein, we report the synthesis of thiazoles and 4-thiazolidinones containing scaffolds (A and B) and study their antiproliferative activity. Four human cancerous cell lines (all of ATCC cell lines) (the breast cancer (MCF-7) cell line, the epithelial cancer (A-549) cell line, the pancreatic cancer (Panc-1) cell line, and the colon cancer (HT-29) cell line) were subjected to an MTT assay to evaluate the newly synthesized thiazoles and 4-thiazolidinones containing motifs. All cells are obtained from The American Type Culture Collection Company, Manassas, Virginia, USA. 

The most effective inhibitory compounds were tested in vitro for EGFR and BRAF^V600E^ targets. Molecular docking analysis evaluated how these molecules attached to the active sites of EGFR and BRAF^V600E^.

## 2. Results and Discussion

Herein, we studied the behavior of cycloalkylidenehydrazinecarbothioamides (**7a**–**f**) toward 2-bromoacetophenone **8** and diethylacetylenedicarboxylate **11**. Cyclocondansation of 2-bromoacetophenone **8** with cycloalkylidenehydrazinecarbothioamides (**7a**–**f**) in absolute ethanol under reflux for three hours afforded the formation of thiazole-containing derivatives **9a**–**f** (Scaffold A) in addition to 2-amino-5-phenyl-3,6-dihydro-2H-1,3,4-thiadiazole (**10**) (Figure 1).

For structure prevalence, we chose derivative **9f** and investigated its spectral data. The IR and ^13^C NMR did not reveal any absorbance or signal of the C=S group. The ^1^H NMR spectrum of **9f** showed a broad signal at δ_H_ 13.13 ppm, corresponding to isatin-NH, and a singlet signal at δ_H_ 6.93, corresponding to thiazole-CH. On the other hand, ^13^C NMR showed characteristic signals at δc 171.25, 168.92, 165.28, 144.11, and 102.41, corresponding to C=O, thiazole-C-2, thiazole C-4, C=N, and thiazole CH, respectively. The structure of **9f** was unambiguously proved by X-ray structure analysis (Figure 3). The ^1^H NMR spectrum of compound **10** showed the thiadiazine-CH_2_ (C-6) and thiadiazine-CH (C-2) at 2.57–2.59 and 4.29, respectively, and two broad signals at δ_H_ 7.39 and 7.88 ppm due to thiadiazine-NH and the amino group, respectively. On the other hand, the ^13^C NMR spectrum of compound **10** showed signals at δc = 35.06, 100.34, and δc = 169.77, which were assigned as thiadiazine-CH_2_, thiadiazine-CH, and thiadiazole C=N, respectively. Moreover, the mass spectrometry exhibited a molecular ion peak at *m*/*z* 221 (M+, 100), which was in agreement with the proposed structure and clearly showed the reaction of thiosemicarbazide **C** after the partial hydrolysis of **7a**–**f** with 2-bromoacetophenone and elimination of HBr and H_2_O followed by aromatization (Figure 2). The single crystal X-ray structure analysis of **10** confirmed that the molecular formula of thiadiazole ring **10** was C_11_H_15_N_3_S, M.wt = 221. The bond length for thiadiazole **10** S1-C1 = 1.7183 (11), S2-C2 = 1.8095 (14), N2-N3 = 1.3879 (15), N2-C1 = 1.3388 (17), N3-C3 = 1.2849 (16) and C2-C3 = 1.2849 (16). The torsion angles for thiadiazole S1-C2-C3-N3 = 43.96 (15), S1-C2-C3-C4 = 141.16 (10), N2-N3-C3-C2 = 2.23 (18), C2-S1-C1-N2 = 24.14 (12), C2-C3-C4-C5 = 9.84 (18) (Figure 4). 

The thiazole derivatives **9a**–**f** (Scaffold A) could be formed according to the proposed mechanism (Figure 2), which starts with the initial conjugate addition of the sulfur lone pair of electrons of compounds **7a**–**f** to the methylene carbon to form salt, then an intramolecular nucleophilic attack of the NH_2_ group on the carbonyl carbon to form intermediate, followed by elimination of the water molecule to yield **9a**–**f**.

The 4-thiazolidinone derivatives **12a**–**f** (Scaffold B) could be formed according to the proposed mechanism (Figure 2), which started with the initial conjugate addition of the sulfur lone pair of electrons of compounds **7a**–**f** to the acetylenic triple bond, followed by intramolecular nucleophilic attack of the NH group on the ester carbonyl carbon in intermediate, followed by elimination of the methanol molecule to yield **12a**–**f**.

The disappearance of C=S, one ester group, and NH_2_ in the spectral data confirmed the cyclocondensation of cycloalkylidenehydrazinecarbothioamides (**7a**–**f**) with diethycetylenedicarboxylate **11**. ^1^H NMR of compound **12a** revealed one triplet and one quartet signal associated with the ethyl group at δ_H_ 1.39 and 4.2, respectively, in addition to a singlet signal at δ_H_ 6.70 and broad one at δ_H_ 7.14, corresponding to vinyl-CH and NH, respectively. Moreover, ^13^ C NMR showed characteristic signals at 176.8, 164.92, 141.6, and 116.58, associated with two C=O, C=N, and vinyl-C, respectively. Furthermore, mass spectroscopy and X-ray analysis confirmed the proposed structure, Figure 5 and Figure 6.

### 2.1. Biology

#### 2.1.1. Cell Viability Assay

The human mammary gland epithelial (MCF-10A) cell line was used to test the viability of new compounds **9a**–**f** and **12a**–**f**. Using the MTT test, the cell viability of compounds **9a**–**f** and **12a**–**f** was evaluated after four days of incubation on MCF-10A cells [25,26]. As shown in Table 1, none of the compounds tested were cytotoxic, and all compounds showed cell viability at 50 µM levels greater than 89%.

#### 2.1.2. Antiproliferative Assay

The antiproliferative effect of compounds **9a**–**f** and **12a**–**f** was evaluated using the MTT assay against four human cancer cell lines: the colon cancer (HT-29) cell line, the pancreatic cancer (Panc-1) cell line, the lung cancer (A-549) cell line, and the breast cancer (MCF-7) cell line [27,28]. Erlotinib was used as a control. Table 1 displays the median inhibitory concentration (IC_50_) and the average inhibitory concentration (GI_50_) against the four cancer cell lines.

In general, compounds **9a**–**f** and **12a**–**f** showed significant antiproliferative activity, with GI_50_ values ranging from 35 nM to 84 nM, compared to that of erlotinib, which had a GI_50_ value of 33 nM. Scaffold A compounds (**9a**–**f**) had GI_50_ values ranging from 35 nM to 78 nM, while Scaffold B compounds (**12a**–**f**) had GI_50_ values ranging from 42 nM to 84 nM, suggesting that Scaffold A compounds are more tolerated than Scaffold B compounds as antiproliferative agents. The most effective derivatives were compounds **9c**, **9f**, **12d**, **12e**, and **12f**, with GI_50_ values ranging from 35 nM to 47 nM, which were 1.1- to 1.4-fold less potent than erlotinib.

Compound **9c** (R = cyclooctylidene, Scaffold A) was the most potent derivative of all synthesized compounds, with a GI_50_ value of 35 nM, comparable to that of erlotinib (GI_50_ = 33 nM). Compound **9c** was even more potent than erlotinib against the breast cancer cell line (MCF-7), with an IC_50_ value of 37 nM, while erlotinib had an IC_50_ value of 40 nM. Replacement of the cyclooctylidene group in compound **9c** with cyclopentyl, as in compound **9a** (R = cyclopentyl, Scaffold A), or cyclohexyl, as in compound **9b** (R = cyclohexyl, Scaffold A), resulted in a marked decrease in antiproliferative action, with GI_50_ values of 78 nM and 64 nM, respectively. Similarly, replacing the cyclooctylidene group with heterocyclic moiety, as in compound **9f** (R = indolin-3-one, Scaffold A), or a polycyclic group, as in compounds **9e** (R = 9H-fluoren, Scaffold A) and **9d** (R = 3,4-dihydronaphthalene, Scaffold A), resulted in a decrease in the antiproliferative activity, with GI_50_ values 44 nM, 50 nM, and 59 nM, respectively. These findings demonstrated that the nature of the substituent at the N-2 of the hydrazinyl moiety of Scaffold A compounds plays an important role in the antiproliferative activity of **9a**–**f**, with activity increasing in the order cyclooctylidene > indolin-3-one > 9H-fluorene > dihydronaphthalen > cyclohexyl > cyclopentyl.

Compounds **12d** (R = 3,4-dihydronaphthalenyl, Scaffold B) and **12e** (R = 9H-fluorenyl, Scaffold B) were ranked second and third in activity, with GI_50_ values of 38 nM and 42 nM, respectively, and were 1.1-fold and 1.2-fold less potent than compound **9c**. For Scaffold B compounds, the nature of the substituent at the N-2 of the hydrazinyl moiety was a bit different than that of Scaffold A compounds, with activity increased in the order of dihydronaphthalen > 9H-fluorene > indolin-3-one > cyclooctylidene > cyclohexylidene > cyclopentylidene. 

Finally, the cyclopentyl derivatives **9a** (R = cyclopentyl, Scaffold A) and **12a** (R = cyclopentyl, Scaffold B) were the least active, with GI_50_ values of 78 nM and 84 nM, respectively, demonstrating that the cyclopentyl group is not favored for antiproliferative action of these types of compounds.

#### 2.1.3. EGFR Inhibitory Assay

Using erlotinib as the reference drug, the most effective derivatives, **9c**, **9f**, **12d**, **12e**, and **12f**, were further explored for their inhibitory action against EGFR as a possible mechanistic target for their antiproliferative action [29]. The IC_50_ values for each compound and for erlotinib are shown in Table 2. Compared to erlotinib (IC_50_ = 80 nM), the evaluated compounds displayed good anti-EGFR efficacy, with IC_50_ values ranging from 86 nM to 100 nM. All of the studied compounds were less effective than erlotinib against EGFR.

The results of this assay are comparable with the results of the antiproliferative assay, in which the most effective antiproliferative agent, compound **9c** (R = cyclooctylidene, Scaffold A), was also the most potent EGFR inhibitor, with an IC_50_ value of 86 nM, compared to erlotinib’s IC_50_ value of 80 nM, being 1.1-fold less potent than erlotinib. Compounds **12d** (R = 3,4-dihydronaphthalen, Scaffold B) and **12e** (R = 9H-fluoren, Scaffold B) ranked second and third in EGFR inhibitory effect, with IC_50_ values of 89 nM and 91 nM, respectively. Finally, compounds **9f** (R = indolin-3-one, Scaffold A) and **12f** (R = indolin-3-one, Scaffold B) had moderate anti-EGFR activity, with IC_50_ values of 97 nM and 100 nM, respectively. These findings indicated that compounds **9c** and **12d** are potential antiproliferative agents with EGFR inhibitory activity.

#### 2.1.4. BRAF^V600E^ Inhibitory Assay

Compounds **9c**, **9f**, **12d**, **12e**, and **12f** were tested for anti-BRAF^V600E^ activity using erlotinib as the control medication [30]. Table 2 displays the IC_50_ values for each compound and for erlotinib. 

Once again, compounds **9c** and **12d**, the most active antiproliferative derivatives, were the most potent BRAF^V600E^ inhibitors, with IC_50_ values of 94 nM and 98 nM, respectively, compared to erlotinib (IC_50_ = 60 nM), being roughly 1.6-fold less potent than erlotinib. The other three derivatives, **9f**, **12e**, and **12f**, displayed weak to moderate efficacy against BRAF^V600E^, with IC_50_ values of 117 nM, 105 nM, and 125 nM, respectively. These findings indicate that the examined compounds require structural modifications in order to yield more effective variants. Moreover, these in vitro experiments revealed that compounds **9c** and **12e** could be effective antiproliferative agents with dual targeting action against EGFR and BRAF^V600E^.

#### 2.1.5. Docking Study

The most effective molecules, **9c**, **9f**, **12d**, **12e**, and **12f**, were selected for further investigation regarding their potential to interact with the active sites of EGFR and BRAF, using erlotinib as a reference compound [31,32]. 

The study aimed to evaluate the efficacy of Scaffold A (compounds **9c,f**) and Scaffold B (compounds **12d,e,f**) as inhibitors of EGFR and BRAF. The results revealed a positive interaction pattern for these molecules within the active sites of EGFR and BRAF, as outlined in Table 3 and Table 4. Notably, among the five test compounds, compound **3c** exhibited the highest docking scores, −6.97 and −7.90 (S; kcal/mol), compared to those of the reference compound erlotinib, which had scores of −7.38 and −8.04, respectively.

The top docking poses of the five test compounds, when compared with that of erlotinib, indicated that these compounds exhibited stability within the active site cavity, with several H-bonds and pi-H hydrophobic interactions with the several residues of amino acids around the active site, as illustrated in Figure 7. Compound **9c** within the active sites of EGFR has two hydrogen bonds with Asp 831 and pi-H hydrophobic interaction with Cys 773, while erlotinib forms two hydrogen bonds with Met 769 and water molecule and pi-H hydrophobic interaction with Lys 721. 

On the other hand, compound **9c** within active sites of BRAF has two hydrogen bonds with Gln 530 and Cys 532 (Figure 8). The order of the docking scores fits with the results of the biochemical tests. Compound **9c** (with R = cyclooctylidene, Scaffold A) emerged with the highest docking score, about 1.1-fold less than that of erlotinib.

Compounds **12d** (with R = 3,4-dihydronaphthalenylidene, Scaffold B) and **12e** (with R = 9H-fluoren, Scaffold B) had the second and third highest docking scores, respectively. Therefore, it was obvious that the stated docking results agreed with the biological findings.

Melting point apparatus (i.e., the Gallenkamp melting point apparatus) was used. Infrared spectra (IR) were performed with Bruker Alpha instruments with a wavelength ranging from 4000 to 600 cm^−1^ as KBr disks. NMR spectra were recorded on a Bruker AM 400 MHz spectrometry using CDCl_3_ as a solvent, with TMS as the internal standard (δ = 0). The data were reported as follows: chemical shift, multiplicity (s = singlet, d = doublet, t = triplet, m = multiplet, and q = quartet). ^13^C NMR, TMS (S=O) was used as the internal solvent, and spectra were observed with complete proton decoupling. Mass spectrometers were recorded on a Finnegan MAT instrument (70 ev, EI-mode). Elemental analyses for C, H, N, and S were obtained using Elmyer 306, and preparative layer chromatography (plc) was carried out using glass plates covered with a 1.0 mm thick silica gel (Merk Pf_254_). 

## 3. Materials and Methods

Cycloalkylidenehydrazinecarbothioamides (**7a**–**f**) were prepared according to reported methods [33].

**Synthesis of thiazole derivatives (9a–f), and 1,3,4-thiadiazole derivatives (10).** A solution of **7a**–**f** (1.0 mmol) and 2-bromoacetophenone (**8**) (1.0 mmol) in absolute ethanol was used as the solvent. The reaction mixture was refluxed for three hours or until the starting solution was fully consumed (the reaction was monitored by TLC analyses). The reaction mixture was filtered, and the precipitate was washed several times with ethanol. The filtrate was evaporated and subjected to chromatographic plates, using toluene-ethylacetate (10:5) as the eluent. The fastest migration zone contained thiazole derivatives (**9a**–**f**), and the slowest zone contained 1,3,4-thiadiazole derivatives (**10**). The isolated products obtained were recrystallized from suitable solvents.

***2-(2-cyclopentylidenehydrazinyl)-4-phenylthiazole* (9a).** Colorless crystals (ethanol) m. p. = 214–216 °C; IR (KBr): υ = 3270 (NH), 3032 (Ar-CH), 2960 (ali-CH_2_), 1624 (C=N), 1478 (Ar-C=C), ^1^H NMR (CDCl_3_): *δ* = 1.77, 1.80, 2.45, 2.48–2.61 (m, 8H, Cyclic CH_2_), 6.62 (s, 1H, thiazole-CH), 7.35–7.40 (m, 5H, Ar-CH), 12.15 (br, 1H, NH), ^13^C NMR (CDCl_3_) *δ* = 24.92, 25.07, 31.07, 33.47 (Ali-CH_2_), 100.21 (thiazole-CH), 125.61, 127.34, 129.62, 130.43 (Ar-CH), 140.66 (C=N), 164.35 (thiazole-C4), 172.75 (thiazole-C2), Ms = *m*/*z* 257 (M^+^, 100), C_14_H_15_N_3_S (257). Anal. Calcd. For C_14_H_15_N_3_S: C, 64.83; H, 6.61; N, 16.20; S, 12.36. Found C, 64.91; H, 6.58; N, 16.32; S, 12.41.

***2-(2-cyclohexylidenehydrazinyl)-4-phenylthiazole* (9b).** Colorless crystals (ethanol) m. p. = 206–208 °C; IR (KBr): υ = 3266 (NH), 3049 (Ar-CH), 2915 (ali-CH_2_), 1609 (C=N), 1476 (Ar-C=C), ^1^H NMR (CDCl_3_): *δ* = 1.50, 1.61, 1.70, 2.32, 2.50–2.71 (m, 10H, Cyclohexyl-CH_2_), 6.61 (s, thiazole-CH), 7.40–7.44, 7.64 (m, 5H, Ar-CH), 12.43 (br, 1H, NH) ^13^C NMR (CDCl_3_) *δ* = 25.28, 26.14, 27.08, 29.42, 35.05 (cyclohexyl-CH_2_), 100.34 (thiazole-CH), 125.59, 127.39, 129.62, 130.40 (Ar-CH), 140.60 (C=N), 164.35 (thiazole-C4), 169.77 (thiazole-C2), Ms = *m*/*z* 271 (M^+^, 100), C_15_H_17_N_3_S (271). Anal. Calcd. For C_14_H_15_N_3_S: C, 66.39; H, 6.31; N, 15.48; S, 11.81. Found C, 66.45; H, 6.44; N, 15.37; S, 11.92. 

***2-(2-cyclooctylidenehydrazinyl)-4-phenylthiazole* (9c).** Colorless crystals (ethanol) m. p. = 198–200 °C; IR (KBr): υ = 3273 (NH), 3051 (Ar-CH), 2914 (Ali-CH_2_) 1601 (C=N), 1476 (Ar-C=C), ^1^H NMR (CDCl_3_): *δ* = 1.48–1.90 (m, 2H, Cyclic CH_2_), 2.41–2.59 (m, 12H, cyclic-CH_2_), 6.62 (s, 1H, thiazole-CH), 7.38–7.67 (m, 5H, Ar-CH), 12.30 (br, 1H, NH), ^13^C NMR (CDCl_3_) *δ* = 24.60, 25.15, 25.33, 26.15, 27.35, 29.85, 36.12 (cyclic-CH_2_), 100.43 (thiazole-CH), 125.60, 127.39, 129.63, 130.38 (Ar-CH), 140.64 (octyl-C=N), 164.79 (thiazole-C4), 169.70 (thiazole-C2), Ms = *m*/*z* 299 (M^+^, 100), C_17_H_21_N_3_S (299). Anal. Calcd. For C_17_H_21_N_3_S: C, 68.19; H, 7.07; N, 14.03; S, 10.71. Found C, 68.22; H, 7.01; N, 14.12; S, 10.82. 

***(E)-2-(2-(3,4-dihydronaphthalen-1(2H)-ylidene)hydrazinyl)-4-phenylthiazole* (9d).** Colorless crystals (ethanol) m.p. = 208–210 °C; IR (KBr): υ = 3025 (NH), 2925 (Ar-CH_2_), 1601 (C=N), 1491 (dihydro-C=C), ^1^H NMR (CDCl_3_): *δ* = 1.94 (2H, cyclic-CH_2_), 1.99, 2.01 (2H, cyclic-CH_2_), 2.75–2.90 (m, 2H, Cyclic CH_2_), 6.69 (s, 1H, thiazole-CH), 7.29–8.03 (m, 9H, Ar-CH), 12.54 (br, 1H, NH), ^13^C NMR (CDCl_3_) *δ* = 21.56, 27.93, 29.39, (cyclic-CH_2_), 100.88 (thiazole-CH), 125.22, 125.65, 126.68, 127.39, 128.90, 129.66, 130.56, 130.73 (Ar-CH), 141.06 (dihydro-C=N), 156.09 (thiazole-C4), 164.81 (thiazole-C2), Ms = *m*/*z* 319 (M^+^, 100), C_19_H_17_N_3_S (319). Anal. Calcd. For C_19_H_17_N_3_S: C, 71.44; H, 5.36; N, 13.16; S, 10.04. Found C, 71.51; H, 5.30; N, 13.22; S, 10.11. 

***2-(2-(9H-fluoren-9-ylidene)hydrazinyl)-4-phenylthiazole* (9e).** Colorless crystals (ethanol) m. p. = 250–252 °C; IR (KBr): υ = 3180 (NH), 3049 (Ar-CH), 1579 (C=N), 1490 (Ar-C=C), ^1^H NMR (CDCl_3_): *δ* =6.84 (s, 1H, thiazole-CH), 7.19, 7.44–7.47 (m, 8H, Ar-CH), 8.9 (br, S, 1H, NH), ^13^C NMR (CDCl_3_) *δ* = 100.28 (thiazole-CH), 120.69, 122.43, 125.87, 128.11, 128.28, 128.36, 129.25, 129.68, 130.54 (Ar-CH), 131.45, 132.57, 136.07 (Ar-C), 142.61 (ylidene-C=N), 164.87 (thiazole-C4), 171.17 (thiazole-C2), Ms = *m*/*z* 353 (M^+^, 100), C_22_H_15_N_3_S (253). Anal. Calcd. For C_19_H_17_N_3_S: C, 74.76; H, 4.28; N, 11.89; S, 9.07. Found C, 74.81; H, 4.21; N, 11.96; S, 9.12. 

***(Z)-3-(((Z)-4-phenylthiazol-2(3H)-ylidene)hydrazineylidene)indolin-2-one (*9f*).*** Colorless crystals (ethanol) m. p. = 270–272 °C; IR (KBr): υ = 3179 (NH), 3026 (Ar-CH), 1673 (C=O), 1614 (C=C), 1460 (Ar-C=C), ^1^H NMR (CDCl_3_): *δ* =6.93 (s, 1H, thiazole-CH), 7.05–7.28 (m, 3H, Ar-CH), 7.34–7.37 (m, 2H, Ar-H), 7.45–7.50 (m, 2H, Ar-H), 7.60–7.62 (m, 2H, Ar-CH), 11.05 (br, S, 1H, NH), 13.13 (br, S, 1H, isatine-NH), ^13^C NMR (CDCl_3_) *δ* = 102.41 (thiazole-CH), 121.19, 122.63, 125.92, 128.41, 128.98, 129.45, 130.14 (Ar-CH), 132.15, 133.17, 136.20 (Ar-C), 144.11 (ylidene-C=N), 165.28 (thiazole-C4), 168.92 (thiazole-C2), 171.25 (C=O), Ms = *m*/*z* 320 (M^+^, 100), C_17_H_12_N_4_OS (320). Anal. Calcd. For C_17_H_12_N_4_OS: C, 63.73; H, 3.78; N, 17.49; S, 10.01, Found C, 63.81; H, 3.85; N, 17.54; S, 10.08. 

***2-Amino-5-phenyl-3,6-dihydro-2H-1,3,4-thiadiazin-2-amine* (10).** Colorless crystals (ethanol) m. p. = 236–238 °C; IR (KBr): υ = 3249–3160 (NH_2_ and NH), 3040 (Ar-CH), 1560 (Ar-C=C), ^1^H NMR (CDCl_3_): *δ* 2.57–2.59 (s, 2H, thiadiazine-CH_2_), 4.29 (s, 1H, thiadiazine-CH), 7.39 (br, s, 1H, NH), 7.88 (br, s, 2H, NH_2_), 7.41–7.64 (m, 5H, Ar-H), ^13^C NMR (CDCl_3_) *δ* = 35.06 (thiadiazine-CH_2_), 100.34 (thiadiazine-CH), 125.59, 127.39, 129.62, 130.40 (Ar-CH), 169.77 (thiadiazine-C=N), Ms = *m*/*z* 221 (M^+^, 100), C_11_H_15_N_3_S (221). Anal. Calcd. For C_11_H_15_N_3_S: C, 59.70; H, 6.83; N, 18.99; S, 14.49. Found C, 59.64; H, 6.75; N, 18.91; S, 14.54.

**Synthesis of (cycloalkylidenehydrazono)-4-oxothiazolidin-5-ylidene)acetates (12a**–**f).** To the solution of diethylacetylenedicarboxylate (1.0 mmol) (**11**) in ethanol, the solution of cycloalkylidenehydrazinecarbothioamides (**7a**–**f**) was added; the mixture was refluxed for four hours until the starting (**7a**–**f**) was fully consumed (TLC monitoring). The reaction mixture was cooled, and the oxothiazolidin-5-ylideneacetates were precipitated. The precipitate was filtered off and recrystallized from ethanol to yield pure crystals from **(12a**–**f)**.

***(E)-ethyl 2-((E)-2-(cyclopentylidenehydrazono)-4-oxothiazolidin-5-ylidene)acetate* (12a).** Colorless crystals (ethanol) m. p. 136–138 °C; IR (KBr): υ = 3410 (NH), 2960 (Ali-CH), 1689 (CO) 1648 (CO), 1582 (C=C), ^1^H NMR (CDCl_3_): *δ* = 1.39 (t, 3H, *J* = 6.8 Hz; CH_3_), 1.55 (s, 4H, Cyclic CH_2_), 1.85 (s, 2H, Cyclic CH_2_), 2.42 (s, 2H, Cyclic CH_2_), 4.20 (q, 2H, *J =* 6.9 Hz; CH_2_), 6.70 (s, 1H, vinyl-CH), 7.14 (s, 1H, NH), ^13^C NMR (CDCl_3_) *δ* = 26.18, 27.86, 31.40, 34.25, 43.60, (cyclic-CH_2_), 116.58 (vinyl-CH), 141.60 (cyclic C=N),164.92 (CO-ester), 176.80 (CO-ketone), Ms = *m*/*z* 281 (M^+^, 100), C_12_H_15_N_3_O_3_S (281). Anal. Calcd. For C_12_H_15_N_3_O_3_S: C, 51.23; H, 5.37; N, 14.94; S, 11.40. Found C, 51.32; H, 5.28; N, 14.86; S, 11.48.

***(E)-ethyl 2-((E)-2-(cyclohexylidenehydrazono)-4-oxothiazolidin-5-ylidene)acetate* (12b).** Colorless crystals (ethanol) m. p. = 168–170 °C; IR (KBr): υ = 3129 (NH), 2936 (Ali-CH), 1691 (CO) 1637 (CO), 1581 (C=C), ^1^H NMR (CDCl_3_): *δ* = 1.30 (t, 3H, *J =* 7.2 Hz; CH_3_), 1.48, 1.87, 2.41, 2.52 (Cyclic CH_2_), 2.69 (Cyclic CH_2_), 4.20 (q, 2H, *J =* 7.1 Hz; CH_2_), 6.78 (s, 1H, vinyl-CH), 7.27 (br, s, NH), ^13^C NMR (CDCl_3_) *δ* = 25.07, 26.26, 27.52, 29.77, 34.72, (cyclic-CH_2_), 116.55 (vinyl-CH), 141.39 (Cyclic C=N), 164.32 (CO-ester), 173.17 (CO-ketone), Ms = *m*/*z* 295 (M^+^, 100), C_13_H_17_N_3_O_3_S (295). Anal. Calcd. For C_13_H_17_N_3_O_3_S: C, 52.87; H, 5.80; N, 14.23; S, 10.85. Found. C, 52.81; H, 5.72; N, 14.28; S, 10.94.

***(E)-ethyl 2-((E)-2-(cyclooctylidenehydrazono)-4-oxothiazolidin-5-ylidene)*acetate (12c).** Colorless crystals (ethanol) m. p. = 146–148 °C; IR (KBr): υ = 3112 (NH), 2925 (Ali-CH), 1680 (CO) 1620 (CO), 1574 (C=C), ^1^H NMR (CDCl_3_): *δ* = 1.33 (t, 3H, *J =* 7.1 Hz; CH_3_), 1.53–2.72 (m, 14Cyclic CH_2_), 4.34 (q, 2H, *J =* 7.1 Hz; CH_2_), 6.81 (s, 1H, vinyl-CH), 7.49 (br, s, 1H, NH), ^13^C NMR (CDCl_3_) *δ* = 23.66, 26.52, 27.17, 29.82, 33.12 (cyclic-CH_2_), 115.15 (vinyl-CH), 140.80 (Cyclic C=N),164.22 (CO-ester), 170.87 (CO-ketone), Ms = *m*/*z* 323 (M^+^, 100), C_15_H_21_N_3_O_3_S (323). Anal. Calcd. For C_15_H_21_N_3_O_3_S: C, 55.71; H, 6.55; N, 12.99; S, 9.91. Found C, 55.82; H, 6.49; N, 12.92; S, 9.98.

***(E)-ethyl 2-((E)-2-(((E)-3,4-dihydronaphthalen-1(2H)-ylidene)hydrazineylidene)-4-oxothiazolidin-5-ylidene)acetate* (12d).** Colorless crystals (ethanol) m. p. = 234–236 °C; IR (KBr): υ = 3155 (NH), 2935 (Ali-CH), 1685 (ester-CO) 1628 (cyclic-CO), 1608 (Ar-C=C), ^1^H NMR (CDCl_3_): *δ* = 1.34 (t, 3H, *J =* 7.2 Hz; CH_3_), 1.52 (m, 2H, Cyclic CH_2_), 1.87 (m, 2H, Cyclic CH_2_), 2.69–2.86 (m, 2H, cyclic-CH_2_), 4.26 (q, 2H, *J =* 7.2 Hz; CH_2_O), 6.90 (s, 1H, vinyl-CH), 7.35–7.52 (m, 4H, Ar-H), 8.29 (br, s, 1H, NH) ^13^C NM R (CDCl_3_) *δ* = 14.07 (CH_3_), 21.96, 27.12, 29.77 (cyclic-CH_2_), 61.19 (CH_2_O), 116.57 (vinyl-CH), 126.35–131.58 (Ar-CH), 141.70 (Cyclic C=N), 164.97 (CO-ester), 172.77 (CO-ketone), Ms = *m*/*z* 343 (M^+^, 100), C_17_H_17_N_3_O_3_S (343). Anal. Calcd. For C_17_H_17_N_3_O_3_S: C, 59.46; H, 4.99; N, 12.24; S, 9.34. Found C, 59.38; H, 4.90; N, 12.18; S, 9.28. 

***(E)-ethyl 2-((E)-2-((9H-fluoren-9-ylidene)hydrazono)-4-oxothiazolidin-5-ylidene)acetate* (12e).** Colorless crystals (ethanol) m. p. 240–242°C; IR (KBr): υ = 3208 (NH), 1717 (ester-CO) 1691 (cyclic-CO), 1609 (Ar-C=C), ^1^H NMR (CDCl_3_): *δ* = 1.31 (t, 3H, *J =* 6.9 Hz; CH_3_), 4.27–4.29 (q, 2H, *J =* 6.9 Hz; CH_2_O), 6.97 (s, 1H, vinyl-CH), 7.10–7.19, 7.35–7.65 (m, 7H, Ar-CH), 8.29 (br, s, 1H, NH) ^13^C NMR (CDCl_3_) *δ* = 14.07 (CH_3_), 61.20 (CH_2_O), 116.42 (vinyl-CH), 126.35–129.4, 131.58–134.12 (Ar-CH), 141.66 (cyclic C=N), 164.35 (CO-ester), 172.90 (CO-ketone), Ms = *m*/*z* 377 (M^+^, 100), C_20_H_15_N_3_O_3_S (377). Anal. Calcd. For C_20_H_15_N_3_O_3_S: C, 63.65; H, 4.01; N, 11.13; S, 8.49. Found C, 63.72; H, 4.08; N, 11.06; S, 8.54. 

***Ethyl (E)-2-((E)-4-oxo-2-(((Z)-2-oxoindolin-3-ylidene)hydrazineylidene)thiazolidin-5-ylidene)-acetate (*12f*).*** Colorless crystals (ethanol) m. p. = 250–252 °C; IR (KBr): υ = 3363 (NH), 1690 (ester-CO) 1598 (cyclic-CO), 1544 (Ar-C=C), ^1^H NMR (CDCl_3_): *δ* = 1.33 (t, 3H, *J* = 7.0 Hz; CH_3_), 4.29 (q, 2H, *J =* 7.0 Hz; CH_2_O), 6.90 (s, 1H, vinyl-CH), 7.19–7.61 (m, 4H, Ar-H), 8.51 (br, s, 1H, NH). ^13^C NMR (CDCl_3_) *δ* = 14.17 (CH_3_), 61.90 (CH_2_O), 116.80 (vinyl-CH), 120.24, 124.10, 128.04, 131.24 (Ar-CH), 141.80 (Cyclic C=N), 164.33 (CO-ester), 166.1, 174.17 (C=O), Ms = *m*/*z* 344 (M^+^, 100), C_15_H_12_N_4_O_4_S (344). Anal. Calcd. For C_15_H_12_N_4_O_4_S: C, 52.32; H, 3.51; N, 16.27; S, 9.31. Found C, 52.41; H, 3.44; N, 16.36; S, 9.38.

## 4. Biology Section

### 4.1. Assay for the Effect of ***9a**–**f*** and ***12a**–**f*** on Cell Viability

To assess the viability of new compounds **9a**–**f** and **12a**–**f**, the human mammary gland epithelial (MCF-10A) cell line was employed. The cell viability of compounds **9a**–**f** and **12a**–**f** was assessed using the MTT test [25,26]. For more details, see the Appendix A. 

### 4.2. Assay for Antiproliferative Activity

The MTT assay was used to assess the antiproliferative activity of compounds **9a**–**f** and **12a**–**f** against four human cancer cell lines, using erlotinib as a control [27,28]. See the Appendix A for more details.

### 4.3. Assay for EGFR Inhibitory Effect

Using erlotinib as a reference medication, the most effective derivatives, **9c**, **9f**, **12d**, **12e** and **12f**, were investigated further for their inhibitory action against EGFR as a potential mechanistic target for their antiproliferative action [29]. Refer to the Appendix A for more details.

### 4.4. Assay for BRAF^V600E^ Inhibitory Effect

The most potent antiproliferative derivatives, **9c**, **9f**, **12d**, **12e** and **12f**, were tested as BRAF^V600E^ inhibitors, using erlotinib as the reference drug [29]. Refer to the Appendix A for more details.

## 5. Protocol of Docking Studies

The automated docking simulation study was performed using Molecular Operating Environment (MOE^®^) version 2014.09. The X-ray crystallographic structure of the target EGFR and BRAF was obtained from the protein data bank (PDB: 1M17, 5JRQ, respectively). The target compounds were constructed in a three-dimensional model using the builder interface of the MOE^®^ program. After checking their structures and the formal charges on atoms by two-dimensional depiction, the following steps were carried out. The target compounds were subjected to a conformational se arch. All conformers were subjected to energy minimization; all the minimizations were performed with MOE until an RMSD gradient of 0.01 Kcal/mole and an RMS distance of 0.1 Å with MMFF94X force-field, and the partial charges were automatically calculated. The protein was prepared for docking studies by adding hydrogen atoms with their standard geometry to the system. The atom’s connection and type were checked for any errors with automatic correction. The selection of the receptor and its atom’s potential were fixed. MOE Alpha Site Finder was used for the active site search in the enzyme structure, using all default items. Dummy atoms were created from the obtained alpha spheres [31,32].

## Data Availability

The data will be provided upon request.

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
