# Peer review of "Synthesis and Antiproliferative Potential of Thiazole and 4-Thiazolidinone Containing Motifs as Dual Inhibitors of EGFR and BRAFV600E"

_molecules, 2023, doi:10.3390/molecules28247951_

Round 1

Reviewer 1 Report

Comments and Suggestions for Authors

The manuscript describes the synthesis and evaluation of thiazole and thiazolidinone derivatives as potential anticancer compounds. The authors highlight the significance of these compounds in the context of tumor-targeted therapies and the importance of finding novel lead compounds with enhanced anticancer efficacy and fewer adverse effects. The study outlines the synthesis process, structural confirmation, and in vitro antiproliferative activity assessment, along with molecular docking analyses targeting EGFR and BRAFV600E. I would recommend this manuscript for acceptance after minor revision.

The results and discussion are generally well-structured and include the synthesis process, structural confirmation techniques, antiproliferative activity evaluation, and molecular docking analysis. However, more details are needed for clarity:

1. Provide reaction conditions (temperatures, reaction times, etc.) for the synthesis of thiazole and thiazolidinone derivatives.

2. Consider improving the visual presentation of chemical structures drawn using ChemDraw.

3. Clarify the cell lines and protocols used for the antiproliferative activity assays.

4. In the supporting information, it's noted that all of the 1H NMR spectra contain solvent impurities, likely due to the challenging purification of the imine compounds. It would be beneficial to integrate all of the product peaks, as this can aid readers in compound identification.

Author Response

Reviewer 1

Comments and Suggestions for Authors

The manuscript describes the synthesis and evaluation of thiazole and thiazolidinone derivatives as potential anticancer compounds. The authors highlight the significance of these compounds in the context of tumor-targeted therapies and the importance of finding novel lead compounds with enhanced anticancer efficacy and fewer adverse effects. The study outlines the synthesis process, structural confirmation, and in vitro antiproliferative activity assessment, along with molecular docking analyses targeting EGFR and BRAFV600E. I would recommend this manuscript for acceptance after minor revision.

The results and discussion are generally well-structured and include the synthesis process, structural confirmation techniques, antiproliferative activity evaluation, and molecular docking analysis. However, more details are needed for clarity:

  1. Provide reaction conditions (temperatures, reaction times, etc.) for the synthesis of thiazole and thiazolidinone derivatives.

Response

The reaction conditions (temperatures, reaction times, etc.) for the synthesis of thiazole and thiazolidinone derivatives have been inserted in scheme 1

  1. Consider improving the visual presentation of chemical structures drawn using ChemDraw.

Response

The visual presentation of chemical structures improved drawn using Chem Draw.

  1. Clarify the cell lines and protocols used for the antiproliferative activity assays.

Response:

We'd like to thank the reviewer. We updated the supplementary file with the methods used for all in vitro experiments that were missing in the first edition. ATCC cell lines (American type cell culture) were used.

  1. In the supporting information, it's noted that all of the 1H NMR spectra contain solvent impurities, likely due to the challenging purification of the imine compounds. It would be beneficial to integrate all of the product peaks, as this can aid readers in compound identification.

Response:

The number of proton has been calculated by nova program

Reviewer 2 Report

Comments and Suggestions for Authors

The article by Hassan et al. is an interesting work describing the synthesis of new thiazole-based analogues with dual activity against EGFR and BRAFV600E. Great antiproliferative activity against multiple cancer cell types, an in-depth SAR analysis and no citotoxicity in healthy cells are the main pros of this work. However, many improvements in contents and style of presentation should be made: introduction should be deployed, synthetic schemes should be implemented, the biological protocols description and conclusions are missing. The following aspects must be checked to improve the quality of the manuscipt. Hence, it is suitable after the following major revisions:

- Implement figure 1 by adding more approved thiazole-based analogues, describing for each of them the clinical use and the approval year. These informations may highlight the important role of this scaffold in medicinal chemistry

- Scheme 1 is incomplete. Some experimental conditions are missing. For example: temperature and timing should be added for each reactions.

- In Scheme 2, remove the numbering for intermediates 13, 14 and 15, since they are not isolated compounds but just a representation of intramolecular rearrangements.

- Figure 3 can be removed and integrated with Scheme 1, since the structures of 9a-f and 12a-f are already depicted there.

- Check the numbering of each pargraph. For example, the Antiproliferative assay paragraph should be 2.1.2 not 2.2.2.

- In table 1, “cell viability” is too generic and is not clear to which cell line it refers to. Change it with MCF-10A.

- The results obtained from the BRAFV600E inhibitory assay should be discussed more closely. In this status, the paragraph is lacking.

- Line 297, change “Reaction of Cycloalkylidenehydrazinecarbothioamides (7a-f) with 2-bromoacetophenone” with “Synthesis of … (9a-f)“ (similiarly to line 367).

- The protocols for the biological assays and the conclusions are missing.

- At the end of line 71, authors state that thiazole-based molecules are “possible inhibitors of several biological target”. However,  apart from cancer-related targets, none of them is described. Few studies discussing, for example, the role in viral or bacterial diseases should be mentioned: Eur J Med Chem. 2023;249:115136. doi: 10.1016/j.ejmech.2023.115136; Curr Top Med Chem. 2021;21(4):257-268. doi: 10.2174/1568026621999201214232458; Chem Biodivers. 2023 Apr;20(4):e202300206. doi: 10.1002/cbdv.202300206; Biotechnol Appl Biochem. 2023 Apr;70(2):659-669. doi: 10.1002/bab.2388; Int J Mol Sci. 2023 Apr 11; 24(8):7092. doi: 10.3390/ijms24087092

 Minor typos to check are:

- IC50 (not IC50) should be written in the same way in the whole manuscript.

- Line 104, remove the round bracket after cell line)

- Add full stop at the end of line 156 and remove it at the end of line 169

- The name of each compound should be in bold in the whole manuscript.

Comments on the Quality of English Language

Minor editing of English language required

Author Response

Reviewer 2

Comments and Suggestions for Authors

The article by Hassan et al. is an interesting work describing the synthesis of new thiazole-based analogues with dual activity against EGFR and BRAFV600E. Great antiproliferative activity against multiple cancer cell types, an in-depth SAR analysis and no cytotoxicity in healthy cells are the main pros of this work. However, many improvements in contents and style of presentation should be made: introduction should be deployed, synthetic schemes should be implemented, the biological protocols description and conclusions are missing. The following aspects must be checked to improve the quality of the manuscript. Hence, it is suitable after the following major revisions:

- Implement figure 1 by adding more approved thiazole-based analogues, describing for each of them the clinical use and the approval year. These information’s may highlight the important role of this scaffold in medicinal chemistry

- Scheme 1 is incomplete. Some experimental conditions are missing. For example: temperature and timing should be added for each reactions.

Response

Conditions temperature and timing has been done

- In Scheme 2, remove the numbering for intermediates 13, 14 and 15, since they are not isolated compounds but just a representation of intramolecular rearrangements.

Response

The number of compounds 13, 14 and 15 has been removed from scheme2 and discussion part.

- Figure 3 can be removed and integrated with Scheme 1, since the structures of 9a-f and 12a-f are already depicted there.

Response

- Figure 3 has been removed

- Check the numbering of each paragraph. For example, the Antiproliferative assay paragraph should be 2.1.2 not 2.2.2.

Response:

the numbering of each paragraph has been checked and corrected

- In table 1, “cell viability” is too generic and is not clear to which cell line it refers to. Change it with MCF-10A.

Response:

In the cell viability assay, we employ a normal human cell line (MCF-10A) to analyze the influence of newly synthesized compounds on the viability of normal cell lines. Any normal cell line can be utilized, and it is still a cell viability experiment, therefore it is more accurate to be cell viability rather than MCF-10A.

- The results obtained from the BRAFV600E inhibitory assay should be discussed more closely. In this status, the paragraph is lacking.

Response:

Done as advised.

- Line 297, change “Reaction of Cycloalkylidenehydrazinecarbothioamides (7a-f) with 2-bromoacetophenone” with “Synthesis of … (9a-f)“ (similarly to line 367).

Response:

Line 297, change has been done

- The protocols for the biological assays and the conclusions are missing.

Response:

Done as advised. The supp. File was updated by the necessary data.

- At the end of line 71, authors state that thiazole-based molecules are “possible inhibitors of several biological target”. However, apart from cancer-related targets, none of them is described. Few studies discussing, for example, the role in viral or bacterial diseases should be mentioned: Eur J Med Chem. 2023;249:115136. doi: 10.1016/j.ejmech.2023.115136; Curr Top Med Chem. 2021;21(4):257-268. doi: 10.2174/1568026621999201214232458; Chem Biodivers. 2023 Apr;20(4):e202300206. doi: 10.1002/cbdv.202300206; Biotechnol Appl Biochem. 2023 Apr;70(2):659-669. doi: 10.1002/bab.2388; Int J Mol Sci. 2023 Apr 11; 24(8):7092. doi: 10.3390/ijms24087092

 Minor typos to check are:

- IC50 (not IC50) should be written in the same way in the whole manuscript.

Done as advised.          

- Line 104, remove the round bracket after cell line)

Done as advised.

- Add full stop at the end of line 156 and remove it at the end of line 169

Done as advised.

- The name of each compound should be in bold in the whole manuscript.

Done as advised.

All corrected comments were highlighted by green colure.

With best regards’

Prof. Dr. Alaa A. Hassan

Round 2

Reviewer 2 Report

Comments and Suggestions for Authors

Authors made the required changes. The article is now suitable for publication.